# Aging and Microglial Response following Systemic Stimulation with *Escherichia coli* in Mice

**DOI:** 10.3390/cells10020279

**Published:** 2021-01-30

**Authors:** Inge C.M. Hoogland, Dunja Westhoff, Joo-Yeon Engelen-Lee, Mercedes Valls Seron, Judith H.M.P. Houben-Weerts, David J. van Westerloo, Tom van der Poll, Willem A. van Gool, Diederik van de Beek

**Affiliations:** 1Department of Neurology, Amsterdam UMC, Academic Medical Center, University of Amsterdam, Amsterdam Neuroscience, Meibergdreef 9, 1105 AZ Amsterdam, The Netherlands; i.c.hoogland@amsterdamumc.nl (I.C.M.H.); d.westhoff@amsterdamumc.nl (D.W.); j.y.lee@amsterdamumc.nl (J.-Y.E.-L.); m.valls.seron@gmail.com (M.V.S.); j.h.p.m.houben-weerts@vu.nl (J.H.M.P.H.-W.); w.a.vangool@amsterdamumc.nl (W.A.v.G.); 2Intensive Care Medicine, Leiden University Medical Centre, Albinusdreef 2, 2333 ZA Leiden, The Netherlands; d.j.van_westerloo@lumc.nl; 3Centre for Experimental and Molecular Medicine, University of Amsterdam, Amsterdam UMC, Amsterdam Infection and Immunity Institute, Meibergdreef 5, 1105 AZ Amsterdam, The Netherlands; t.vanderpoll@amsterdamumc.nl

**Keywords:** microglia, microglial activation, systemic infection, *Escherichia coli*, neuro-inflammation, ageing, mouse model

## Abstract

Systemic infection is an important risk factor for the development cognitive impairment and neurodegeneration in older people. Animal experiments show that systemic challenges with live bacteria cause a neuro-inflammatory response, but the effect of age on this response in these models is unknown. Young (2 months) and middle-aged mice (13–14 months) were intraperitoneally challenged with live *Escherichia coli* (*E. coli*) or saline. The mice were sacrificed at 2, 3 and 7 days after inoculation; for all time points, the mice were treated with ceftriaxone (an antimicrobial drug) at 12 and 24 h after inoculation. Microglial response was monitored by immunohistochemical staining with an ionized calcium-binding adaptor molecule 1 (Iba-1) antibody and flow cytometry, and inflammatory response by mRNA expression of pro- and anti-inflammatory mediators. We observed an increased microglial cell number and moderate morphologically activated microglial cells in middle-aged mice, as compared to young mice, after intraperitoneal challenge with live *E. coli*. Flow cytometry of microglial cells showed higher CD45 and CD11b expressions in middle-aged infected mice compared to young infected mice. The brain expression levels of pro-inflammatory genes were higher in middle-aged than in young infected mice, while middle-aged infected mice had similar expression levels of these genes in the systemic compartment. We conclude that systemic challenge with live bacteria causes an age-dependent neuro-inflammatory and microglial response. Our data show signs of an age-dependent disconnection of the inflammatory transcriptional signature between the brain and the systemic compartment.

## 1. Introduction

Delirium is the most common complication among hospitalized older people, aged 65 years and above, and has been associated with long-term risks of death, institutionalization, and dementia, independently of important confounders [1]. The cause of delirium is typically multifactorial [2], but the pathogenesis of delirium remains poorly understood. Evidence from animal and human studies suggests that systemic inflammation, age and microglial cells play a crucial role [3,4]. Animal models of sepsis have confirmed central nervous system (CNS) inflammation, microglial cell activation and neuronal death [5,6]. Human studies show that systemic inflammation promotes cognitive decline and neurodegenerative disease [7,8]. In light of the recent pandemic, COVID-19 is associated with a severe innate immune response and the production of pro-inflammatory cytokines [9,10]. 

Additionally, age-related morphological changes, elevated expressions of several pro-inflammatory genes and reduced expressions of neuroprotective factors in microglial cells are well-documented, and are referred to as microglial priming [11,12]. Priming makes microglial cells susceptible to secondary stimuli, such as a systemic infection, which can trigger an exaggerated inflammatory response [13,14]. This is also relevant in COVID-19 survivors, considering the COVID-19 virus preferentially targets patients with advanced age. One potential contributor to this protracted and uncontrolled neuro-inflammation during aging is the impaired regulatory signaling of microglial cells [12,15]. The critical roles of CX3CR1, CD200 and CD47 in microglial regulation and the prevention of neurotoxicity have been repeatedly demonstrated in various in vivo models of neuro-inflammation [16,17,18]. In this regard, the Toll-like receptor (TLR) signaling cascade might be relevant, as microglial activation is associated with TLR upregulation, and the TLR signaling cascade can be interrupted by negative regulators at any step in the cascade [19,20,21]. 

The majority of these data is derived from experimental animal studies using (peripheral) systemic inflammatory challenge(s) with lipopolysaccharide (LPS). However, the clinical relevance of using LPS as a stimulus to study mechanisms connecting systemic inflammation, ageing and microglial cell response is debatable. In a systematic review of the literature, we describe how there are distinct differences in the microglial activation and neuro-inflammatory responses between systemic infections with peripheral challenge with LPS and peripheral challenge with live or heat-killed bacteria, whereby the microglial activation and neuro-inflammatory response is less profound in the latter [22]. 

In 2018, we introduced an animal model that closely simulates the clinical situation, whereby systemic challenge with live *Escherichia coli* (*E. coli*) induces microglial activation 3 days after inoculation in young adult mice [23]. The objective of this study is to compare microglial cell response, inflammatory markers and microglial activation inhibitory regulators in the brains of young and middle-aged mice, using this clinically relevant animal model. 

## 2. Materials and Methods

### 2.1. Animals

In total, 78 2-month (young) and 81 13- to 14-month (middle-aged) male specific pathogen-free wild type C57BL/6 mice were purchased from Charles River (Maastricht, The Netherlands) and maintained at the animal care facility of the Academic Medical Centre (University of Amsterdam). The mice were housed in groups of 2 to 6, in individually ventilated cages (IVC), for at least 2 weeks before testing. The ambient temperature was 19–24 °C with 40–70% humidity. According to national guidelines, food and water were available ad libitum and a 12:12 h light–dark cycle was retained. All experiments were approved by the Institutional Animal Care and Use Committee of the Academic Medical Centre (Amsterdam, The Netherlands). 

### 2.2. Bacteria

*E. coli* (K1:O18) was cultured in Luria–Bertani medium (LB) at 37 °C to the midlog phase in 1 h and 45 min. The quantity of bacteria in the culture was estimated by measuring the *A*600 in a spectrophotometer. *E. coli* bacteria were harvested by centrifugation at 3000 rpm for 10 min and washed twice with pyrogen-free sterile isotonic saline. The bacteria were diluted to a final concentration of 1 × 10^4^ colony-forming units (CFUs)/200 μL. Serial ten-fold dilutions of the final bacterial inoculum were plated on blood agar plates and incubated overnight at 37 °C to verify the quantities of viable bacteria injected.

### 2.3. Experimental Procedures

The mice (*n* = 30–35 per time point) were given a single intraperitoneal injection of 1 × 10^4^ CFU in 200 μL (range 1.0 × 10^4^–1.8 × 10^4^ CFU). Because this is a lethal dose of *E. coli*, causing mortality in mice around 20–22 h after injection, we treated all mice intraperitoneally with ceftriaxone, an antimicrobial drug (Fresenius Kabi, Den Bosch, the Netherlands), 12 and 24 h after *E. coli* injection in a dose of 20 mg/kg. In total, 64 control mice (*n* = 6–8 per time point) received 200 μL of pyrogen-free sterile isotonic saline via intraperitoneal injection; 40 of the control mice were also treated with ceftriaxone. Inoculations were performed in 3 different rounds and mice were sacrificed at 2, 3 and 7 days after inoculation. Earlier, we postulated that we would find microglial cell activation 3 days after inoculation in young adult mice [23]. In this study we added the 2- and 7-day timepoints to investigate if microglial activation in middle-aged mice might evolve earlier and/or might persist longer as compared to young mice. An overview of experimental rounds, group size and time points is illustrated in Table A1 of Appendix A.

The weight of all mice was monitored before inoculation, before administration of ceftriaxone/saline and before sacrifice. For the 7-day time point, mice were weighted once a day after the last ceftriaxone injection. Weight loss was considered a quantitative and global measurement for the degree of sickness. Additionally, sickness behavior, such as reduced grooming, reduced locomotor activity, social disinterest and pilo-erection, was monitored during the experiments.

Mice were anesthetized by intraperitoneal injection of ketamine (190 mg/kg, Eurovet Animal Health, Bladel, The Netherlands) and medetomidine (0.3 mg/kg, Pfizer Animal Health, Capelle aan den IJssel, the Netherlands). Body weight was assessed, after which cardiac puncture for blood collection performed. Blood was collected in sterile tubes containing ethylenediaminetetraacetic acid (EDTA) and stored on ice. The abdomen was opened and the vena cava was severed to obtain exsanguination. Subsequently, cerebral spinal fluid (CSF) was collected, by puncturing of the cisterna magna, in sterile tubes and stored on ice. Thereafter the thorax was opened and perfusion of the organs with sterile phosphate buffered saline (PBS) was performed via the left cardiac ventricle (approximately 20 mL PBS in 5 min). Next the spleen, the median lobe of the liver and the brain were harvested. The spleen and median lobe of the liver were taken up in 20% weight per volume sterile saline. The right hemisphere of the brain was either suspended in 10% buffered formalin and embedded in paraffin for histopathology or taken up in 20% weight per volume sterile saline. The left hemisphere of the brain was either suspended in 5 mL of Hibernate-A medium (Invitrogen, Cat. A12475-01) and stored at 4 °C (for isolating microglial cells for flowcytometry) or taken up in 20% weight per volume sterile saline (for RNA extraction and real time PCR). The organs suspended in sterile saline were put on ice and were disrupted with a tissue homogenizer. Directly after homogenizing the organs, 50 μL of the tissue homogenate was suspended in 350 μL RA1 lysis buffer (Kit content of NucleoSpin^®^ RNAII, Macherey-Nagel, Cat. 740955) and stored at 80 °C for messenger ribonucleic acid (mRNA) isolation. CSF was diluted 1:100 in sterile saline because of the low volume. Serial ten-fold dilutions of blood, CSF, liver and spleen homogenates were plated on blood agar plates and the bacteria were allowed to grow overnight at 37 °C. For the exact group sizes for the different analysis techniques, we refer to the figure legends in the results section and Appendix A. 

### 2.4. Isolating Microglia for Flowcytometry

The overnight storage of brain tissue for flow cytometry was necessary because we lacked the manpower to process all the tissue on the same day as we sacrificed the animals. The quality of microglial cells was tested in other experiments and the overnight storage had no influence on the results (data shown in Appendix A, Figure A1). After overnight storage in Hybernate-A medium at 4 °C, left hemispheres (approximately weighing 250 mg) were meshed through a 70 μm cell strainer (nylon, BD Falcon, Cat. 352350) in a glucose–potassium–sodium buffer (GKN-BSA; 8 g/L NaCl, 0.4 g/L KCl, 1.77 g/L Na_2_HPO_4_.2H_2_O, 0.69 g/L NaH_2_PO_4_.H_2_O, 2 g/L D-(1)-glucose, pH 7.4) with 0.3% bovine serum albumin (Roche, Cat. 10735108001) and collected in 50 ml tubes. After centrifuging (1400 rpm, 7 min, 4 °C), cell pellets were suspended in 1 mL enzyme buffer (4 g/L MgCl_2_, 2.55 g/L CaCl_2_, 3.73 g/L KCl, and 8.95 g/L NaCl, pH 6–7), followed by enzymatic digestion in collagenase type I (370 units, Worthington, Cat. 9001-12-1) and DNase I (10 mg/mL, Roche, Cat. 1284932) for 45 min, at 37 °C, while shaking. After enzymatic dissociation, cells were washed with GKN-BSA buffer and incubated for 2 min on ice in 2 mL cold erythrocyte lysis buffer (8.3 g/L NH_4_Cl, 1 g/L KHCO_3_ and 0.03 g/L EDTA, pH 7.4). Subsequently, cells were washed and resuspended in 20 mL Percoll (GE healthcare, Cat. 17-0891-01) of *ρ* = 1.03, then underlain with 10 mL Percoll of *ρ* = 1.095 and overlain with 5 mL of GKN-BSA buffer. The tubes were centrifuged for 35 min at 1200× *g* and 20 °C, with slow acceleration and no break. The myelin layer on the top of the *ρ* = 1.03 phase was discarded and the cells were collected from the interface between *ρ* = 1.095 and *ρ* = 1.03 Percoll. Next the cells were washed and counted with a Coulter counter (Beckman Coulter, Z2). Approximately 1 × 10^5^ cells from every sample were transferred into separate polystyrene-coated round-bottomed 5 mL tubes (BD Falcon, Cat. 352008). Depending on cell numbers, a portion of every sample was put in a pool tube, which subsequently was divided over 5 tubes for a blank sample and single stainings; every tube contained approximately 1 × 10^5^ cells.

The cells were stained in a total volume of 200 μL using antibodies with the following specificities: rat anti-mouse cluster of differentiation (CD) 11b (immunoglobulin (Ig)G2b κ, clone M1/70, labeled with phycoerythrin (PE), 1:200, BD Pharmigen, Cat. 557397) and rat anti-mouse CD45 (IgG2b κ, clone 30-F11, labeled with allophycocyanin (APC), 1:500, eBioscience, Cat. 17-0451). To block specific binding, normal mouse serum (1:10) and anti-mouse CD16/CD32 (Alias Fcγ III/II receptor, IgG2b κ, clone 2.4G2, 1:100, BD Pharmigen, Cat. 553142) were added. Cells were incubated with antibodies for 30 min on ice in polystyrene-coated round-bottomed 5 mL tubes. About 10 min prior to fixation, 2.5 μL of 7-amino-actinomycin D (7-AAD, labeled with peridinin-chlorophyll-protein (PerCP), 1:80 concentration, BD Pharmigen, Cat. 559925) was added per sample. After staining, the cells were washed and fixated in 2% paraformaldehyde for 10 min on ice. The cells were washed and resuspended in 200 μL GKN-BSA buffer. As stated above all mice brains were perfused with PBS to prevent the interference of circulating blood myeloid cells. In earlier experiments we found no evidence of the presence of infiltrated myeloid cells (data shown in Appendix A
Figure A2). Therefore, all CD45- and CD11b-positive cells could be defined as microglial cells and were selected for flow cytometric analysis. Flow cytometric analysis was performed on a FACSCalibur machine (BD) and data were analyzed using FlowJo software version 7.6.1.

### 2.5. Immunohistochemistry of Murine Brain

Histopathology was performed on the right brain hemisphere fixed in 4% paraformaldehyde and paraffin-embedded. Coronal 5 μm-thick sections were cut and stained with hematoxylin and eosin (HE) according to standard procedures. Tissue sections were stained for ionized calcium-binding adaptor molecule 1 (Iba-1, rabbit polyclonal, 1:2000, Wako Pure Chemical Industries, Cat. 019-19741) using immunohistochemical procedures, as described previously [23]. Luxol fast blue–Periodic acid–Schiff–hematoxylin staining was used to discriminate between white and gray matter, and was compared to microglial staining to select different brain regions for analysis. Finally, slides were dehydrated and coverslipped with Pertex.

### 2.6. Quantification of Immunohistochemistry Images

Stained slides were scanned with a D. Sight fluo (A. Menarini, Florence, Italy) at 20x magnification. Two square millimeters of each digital image of cortex, hippocampus, thalamus and caudate nucleus were selected by a pathologist (JYEL) for each experimental group (NaCl: *n* = 6; NaCl + Ceft: *n* = 9; *E. coli* + Ceft 2, 3 and 7d: *n* = 6). Microglial cell bodies were manually counted by one observer, who was blinded for all mice characteristics (ICMH).

### 2.7. Definition of Microglial Activation

Microglial cells were defined as activated based on the following criteria: (1) morphologic characteristics based on immunohistochemical staining; (2) significant increase in number, quantified with immunohistochemical staining, (3) significant increase in expression of a microglial marker, quantified with flow cytometry (expression of CD45 and/or CD11b). When 3 of the stated parameters were positive, microglial cells were defined as activated. When 1 or 2 parameters were positive, microglial cells were defined as moderately activated, and when all parameters were negative, microglial cells were defined as inactive. 

### 2.8. RNA Extraction and Real Time PCR

Total ribonucleic acid (RNA) was extracted from murine brain and spleen homogenates using the nucleospin II extraction kit (Macharey-Nagel GmbH, Duren, Germany). The concentration of the RNA was measured using Nanodrop Spectrophotometer (Nanodrop Technologies, Pittsfield, MA, USA) and the purity was assessed by the ratio of absorbance at 260 and 280 nm. RNA purity was within the range of 2.0–2.1. Complementary deoxyribonucleic acid (cDNA) was synthesized from equal amounts of RNA using Iscript^TM^ according to the manufacturer’s protocol (Biorad Laboratories, Hercules, CA, USA). Gene-specific analysis by real-time polymerase chain reaction (PCR) was performed using an iCycler MyiQTM system with Bio-Rad iQ^TM^SYBRGreenSupermix (Biorad Laboratories, Hercules, USA). Expression levels were normalized to reference gene non-Pit-Oct-Unc (POU)-domain containing octamer binding protein (NoNo). Primer sequences are depicted in Appendix A, Table A2. A negative control without the reverse transcriptase was also used. Data were analyzed using the Bio-Rad MyiQ Optical system software version 1.0 and expression data were calculated using the delta cycle threshold (deltaCt) method. 

We investigated the inflammatory response in brain homogenate by measuring the mRNA expression of general pro-inflammatory mediators (tumor necrosis factor alpha (TNF-α), interleukin 1 beta (IL-1β), interleukin 6 (IL-6), interleukin (IL-12), high-mobility group 1 (HMGB1) and complement component 1, subcomponent q (C1q)) and immune regulatory mediators (monocyte chemotactic protein 1 (MCP-1), macrophage colony-stimulating factor (M-CSF) and transforming growth factor beta (TGF-β)). Additionally, we examined myeloid cell markers Iba-1 and CD11b, and specific microglial genes with anti-inflammatory effects (fractalkine (CX3CL1) and its receptor (CX3CR1), CD200 and its receptor CD200R, and CD47 and its receptor signal regulatory protein α (SIRP-α)) by measuring the mRNA expression in the brain homogenate. Lastly, we examined components of the Toll-like receptor (TLR) signaling cascade by measuring the mRNA expression in the brain homogenate of activators of the TLR signaling cascade TLR-2 and mitogen-activated protein kinase 1 (MAPK1), and inhibitors of the TLR signaling cascade suppressor of cytokine signaling 1 (*SOCS1*) and deubiquitinase protein A20 (A20) [19]. In the systemic compartment we investigated inflammatory response in spleen homogenate by measuring the mRNA expression of TNF-α, IL-1β, HMGB1, IL-6, IL-12, M-CSF, MCP-1 and TGF-β. 

### 2.9. Statistical Analysis

A parametric or non-parametric two-way ANOVA (factors: age and time point) was conducted on immunohistochemistry and mRNA expression data, depending on sample size and the distribution of the data. For non-parametric ANOVAs, ranked transformation of data was performed. Subsequently, continuous variables were tested with post-hoc Mann–Whitney U tests or Student *t*-tests. Flowcytometry for every different time point was done on a different day. Because laser characteristics vary per day, geometric mean fluorescence intensity (geoMFI) is not comparable between the different time points. Therefore, Mann–Whitney U tests were done on flow cytometry data. The assumption of normality was tested with the Levene’s and Shapiro–Wilk tests. Statistical significance was set to *p* ≤ 0.05. Because of the exploratory nature of this study, correction for multiple comparisons was not applied. All statistic tests were done using IBM SPSS Statistics (version 26.0). For a visual presentation of quantitative results, GraphPad Prism was used (GraphPad Software, version 6.07, La Jolla, CA, USA).

## 3. Results

### 3.1. Description of Mouse Models

At a time point 12 h after inoculation, all *E. coli*-infected mice showed signs of sickness; middle-aged mice showed less movement and grooming, but young mice lost more weight (Appendix A
Figure A3). Ceftriaxone treatment was administrated at 12 h after inoculation. Behavior returned to normal at 34 h after inoculation in young mice and at 60 h after inoculation in middle-aged mice, with the exception of six middle-aged mice; these sic *E. coli*-infected middle-aged mice died before the predefined time point. None of the *E. coli*-infected young mice died during the experiments. At day 6, the weight difference of *E. coli*-infected middle-aged mice was recuperated as compared with controls, while the weight difference of *E. coli*-infected young mice differed from the control group (Appendix A, Figure A1). At 2 and 3d after inoculation, CFUs in *E. coli*-infected young mice were positive for spleen and liver, and in *E. coli*-infected middle-aged mice for blood, spleen and liver. All CSF cultures were negative. At day 7, all cultures were negative (Appendix A, Figure A4). 

### 3.2. Microglial Cell Response

In young mice, immunohistochemistry showed an increased cell number in the caudate nucleus as compared to controls at day 2 post-inoculation (median 135 cells/2 mm^2^; interquartile range (IQR) 118–141 vs. 107 cells/2 mm^2^ IQR 100–118 controls; *p* = 0.005; Figure 1), but no morphological changes (data not shown). At day 3 immunohistochemistry showed continued increased cell number in the caudate nucleus (median 126 cells/2 mm^2^ (IQR 122–141) vs. 107 cells/2 mm^2^ (IQR 100–118); *p* = 0.003) and no morphological changes (Figure 2 and Figure 3). At day 7, there was an increased cell number in the cortex (median 123 (IQR 113–119) vs. 98 (IQR 91–121); *p* = 0.04), hippocampus (median 120 (IQR 112–124) vs. 102 (IQR 94–109); *p* = 0.009), thalamus (median 114 (IQR 113–119) vs. 91 (IQR 87–111); *p* = 0.001) and the caudate nucleus (median 127 (IQR 120–135) vs. 107 (IQR 100–118); *p* = 0.008; Figure 1). Additionally, the flow cytometry of isolated microglial cells showed an increased expression of CD45 (*p* = 0.008) and CD11b (*p* = 0.008) (Figure 4). Importantly, at day 7 there was no morphological activation of microglial cells in young mice (Figure 2 and Figure 3).

In middle-aged mice, immunohistochemistry showed an increased cell number in the caudate nucleus as compared to controls at day 2 (median 116 cells/2 mm^2^ (IQR 111–122) vs. 107 cells/2 mm^2^ (IQR 97–115); *p* = 0.05; Figure 1), but no morphological changes. Flow cytometry of isolated microglial cells showed increases in cell size (*p* = 0.04) and expression of CD11b (*p* = 0.04) in the infected groups compared to controls (Figure 4). At day 3, immunohistochemistry showed a continued increase in cell number in the caudate nucleus (median 133 cells/2 mm^2^ (IQR 118–142) vs. 102 cells/2 mm^2^ (IQR 97–115); *p* = 0.002; Figure 1) and moderate morphologically activated microglial cells in the hippocampus and thalamus (Figure 2 and Figure 3). Flow cytometry showed an increase in CD11b expression in the infected groups compared to controls (*p* = 0.04; Figure 3). At day 7, immunohistochemistry showed an increased cell number in the cortex (median 132 (IQR 122–143) vs. 113 (IQR 102–128); *p* = 0.02), hippocampus (median 145 (IQR 134–157) vs. 115 (IQR 104–121); *p* = 0.0006), thalamus (median 133 (IQR 121–145) vs. 107 (IQR 99–114); *p* = 0.0004) and the caudate nucleus (median 134 (IQR 124–148) vs. 102 (IQR 97–115); *p* = 0.0003). Flow cytometry showed increased expressions of CD45 (*p* = 0.01) and CD11b (*p* = 0.008) (Figure 3). The morphological moderate activation of microglial cells was still present at day 7 in the hippocampus and thalamus, but was less profound than at day 3 (Figure 2 and Figure 3).

Immunohistochemistry showed no differences in cell number or morphology between control young and middle-age mice in the different brain regions. There was a higher cell number in the middle-age infected mice thalamus at all time points (day 2, median 109 (IQR 104–121) vs. 99 (IQR 90–104); *p* = 0.03; day 3, median 111 (IQR 106–134) vs. 94 (IQR 80–101); *p* = 0.03; day 7, median 133 (IQR 121–145) vs. 114 (IQR 113–119); *p* = 0.01) compared to young infected mice (Figure 1). At day 7, flow cytometry showed bigger cell sizes in young infected mice (*p* = 0.008). The expressions of CD45 and CD11b were increased for all time points in middle-aged infected mice compared to young infected mice (Figure 4). Morphological microglial activation was not seen in young infected mice, and was only moderate at the 3d and 7d time points in the thalamus and hippocampus in middle-aged infected mice (Figure 2 and Figure 3). 

### 3.3. Inflammatory Mediators in Brain

The analysis of pro-inflammatory mediators (Figure 5) in young mice showed the upregulation of brain mRNA expression of HMGB-1 at day 3 in young mice as compared to controls (*p* = 0.005). The expressions of IL-6 and C1q were decreased at day 7 as compared to controls (respectively (resp.) *p* = 0.03; *p* = 0.01). In middle-aged mice, brain mRNA expression levels of IL-1β were increased at days 2 and 3 as compared to controls (resp. *p* = 0.05; *p* = 0.005), and C1q expression levels were increased at day 2 (*p* = 0.003). The expression of IL-6 was decreased at all time points compared to controls (resp. *p* = 0.009, *p* = 0.002 and *p* = 0.03). When the brain mRNA expressions of pro-inflammatory mediators were compared between young and middle-aged control mice, IL-6 expression was higher in middle-aged control mice (*p* = 0.03). Comparing young and middle-aged infected mice, there were higher expressions of HMGB-1 at day 2 (*p* = 0.004), TNF-α and IL-1β at the 3d time point (resp. *p* = 0.02, *p* = 0.008), and TNF-α and C1q at the 7d time point (resp. *p* = 0.008, *p* = 0.0001) in middle-aged infected mice. For IL-1β, HMGB1, IL-12 and C1q, there was a significant interaction for age and time point, indicating a different course of expression in time for the different age groups. The most notable was IL-1β, the expression of which in middle-aged infected mice increased over time with a maximum expression on day 3, and it returned to baseline at day 7, while in young infected mice there was no change in its expression over time.

The analysis of immune regulatory mediators (Figure 6) showed that brain mRNA expression of MCP-1 was increased at day 2 in young infected mice, as compared to controls (*p* = 0.002). Expressions of MCP-1 and M-CSF were decreased at day 3 as compared to controls (resp. *p* = 0.02; *p* = 0.01). In middle-aged mice, brain mRNA expression levels of MCP-1 were increased at day 2 and 3 as compared to controls (resp. *p* = 0.0008; *p* = 0.01), and expression of M-CSF was decreased at day 3 (*p* = 0.01) and subsequently increased at day 7 (*p* = 0.007). When the brain mRNA expressions of immune regulatory mediators are compared between young and middle-aged control mice, there are no differences. Comparing young and middle-aged infected mice, there was higher expression of MCP-1 at day 3 and day 7 (resp. *p* = 0.002, *p* = 0.02) in middle-aged infected mice. For both M-CSF and MCP-1, there was a significant interaction between age and time point, expressed by an increase in MCP-1 on day 3 and an increase in M-CSF on day 7 in middle-aged infected mice, while in young infected mice the expressions of both cytokines returned to baseline at these time points.

The analysis of the mRNA expression of myeloid cell marker Iba-1 (Figure 7) showed the same trend of expression per time point in young and middle-aged mice compared to their controls. There was no difference in the expression of Iba-1 between young and middle-aged control mice. Comparing young and middle-aged infected mice, there was significantly more expression of Iba-1 at day 3 and day 7 in middle-aged infected mice (resp. *p* < 0.0001, *p* = 0.01), and there was a significant interaction between age and time point, expressed by a less profound decrease in expression over time in middle-aged infected mice.

The analysis of specific microglial genes with anti-inflammatory effects (Figure 8) showed decreased brain mRNA expression of CX3CR1 at day 3 and CD47 at day 7 in young mice as compared to controls (resp. *p* = 0.03, *p* = 0.005). In middle-aged mice, mRNA expressions of CX3CR1, CD200R and CD47 were increased at day 2 and day 3 as compared to controls (*p*’s ranging from 0.03 to *p* < 0.0001). Comparing young and middle-aged infected mice, there were increased expressions of CX3CL1, CD200R and CD47 at day 2 (resp. *p* = 0.008, *p* = 0.0002, *p* = 0.002) and a higher expression of CX3CR1 at day 3 (*p* = 0.01) in middle-aged mice. For CX3CR1, CX3CL1, CD200R and CD47, there was a significant interaction between age and time point, expressed by a more profound increase in expression at days 2 and 3 in middle-aged infected mice. 

The expression analysis of the TLR signaling cascade components (Figure 9) in young mice showed a decrease in expression of MAPK1 and A20 at day 7 as compared to controls (resp. *p* < 0.0001, *p* = 0.02). In middle-aged mice, the mRNA expressions of TLR-2 and MAPK1 were increased at day 3 as compared to controls (resp. *p* < 0.0001, *p* < 0.0001), and the expression of MAPK1 remained higher at day 7 (*p* = 0.05). Comparing young and middle-aged infected mice, there was higher expression of TLR-2 at day 3 in middle-aged infected mice (*p* < 0.0001). For TLR-2, MAPK1 and A20, there was a significant interaction between age and time point, expressed by a more profound increase in expression at day 3 in middle-aged infected mice for all parameters. 

The expression analyses for TFG-β, CD11b, CD200, SIRP-α and SOCS1 showed no differences between young and middle-aged control or infected mice, and no significant interaction between age and time was found (graphs are shown in Appendix A, Figure A5).

### 3.4. Inflammatory Mediators in Spleen

In the spleen, mRNA expression in young infected mice was increased for IL-1β at day 2 (*p* = 0.002), and for TNF-α, IL-6, HMGB-1 and MCP-1 at day 7, as compared to controls (resp. *p* < 0.0001, *p* < 0.0001, *p* < 0.0001, *p* < 0.0001). There was a significant decrease in the mRNA expression of IL-12 at day 2 (*p* = 0,04), IL-12, HMGB-1 and M-CSF at day 3 (resp. *p* = 0.009, *p* = 0.0008, *p* = 0.02) and IL-12 and M-CSF at day 7 (resp. *p* < 0.0001, *p* = 0.001). In middle-aged mice, mRNA expression in the spleen was increased for TNF-α, IL-6 and HMGB-1 at day 7 as compared to the controls (resp. *p* = 0.07, *p* < 0.0001, *p* < 0.0001). Expression was significantly decreased for HMGB-1 and TGF-β at day 2 (resp. *p* = 0.03, *p* = 0.004), IL-12, HMGB-1, M-CSF and TGF-β at day 3 (resp. *p* = 0.03, *p* = 0.003, *p* = 0.001, *p* = 0.02) and IL1-β, IL-12, M-CSF and TGF-β at day 7 (resp. *p* = 0.01, *p* < 0.0001, *p* = 0.0003, *p* = 0.01), compared to controls. Comparing young and middle-aged infected mice, there was a higher expression of M-CSF in middle-aged control mice and in middle-aged infected mice at day 2 (resp. *p* = 0.01, *p* = 0.03). There was no significant interaction between age and time point for all parameters, indicating a similar course of expression over time for the different age groups for all parameters measured in the spleen homogenate (Figure 10).

## 4. Discussion

Our experiments show distinct differences in microglial cell number and the brain inflammatory response between young and middle-aged mice after intraperitoneal challenge with live *E. coli*. In middle-aged infected mice, we observed increased microglial cell numbers in the thalamus, and morphological moderate activated microglial cells, which was most profound in the thalamus and hippocampus, compared to young infected mice. Our findings are in line with the hypothesis that a systemic infection may trigger a more pro-inflammatory response in the brain during aging. Previous experiments showed that peripheral inflammatory stimuli cause microglial activation, but a systemic challenge in these studies was performed with supranatural doses of lipopolysaccharides (LPS), which limits external validity to the clinical conditions in humans [22]. We previously showed that a systemic challenge with live *E. coli* causes a neuro-inflammatory response, but this response occurs at a later time point and is less vigorous as compared to LPS stimulation [23]. We now show that systemic challenge with live bacteria causes an ageing-dependent microglial response that remains for at least one week after systemic infection.

We found region-specific differences in microglial cell count and morphological activation between middle-aged and young infected mice, which was most profound in the thalamus. There were no differences between middle-aged and young control mice. Immunohistochemistry showed an increase in microglia cell number in the thalamus at all time points, and morphological moderate activated microglial cells in the thalamus and hippocampus 3 days and 7 days after infection in the middle-aged infected mice. These differences were not seen in the cortex or caudate nucleus. One study compared the activation of microglial cells with immunohistochemistry in young and old rodents after peripheral infection [24]. This study showed significantly greater microglial activation in 12-month-old rats compared to 2-month-old rats in the hippocampal CA1 region, 3 weeks after intradermal inoculation of heat-killed *Mycobacterium butyricum*. Regional differences in the neuro-inflammatory response after systemic injection of LPS have been described. One study using young mice observed Iba1-positive microglial cells to be increased in the striatum, medial septum, frontal cortex, and hippocampus after LPS treatment [25]. Microglia display a regional heterogeneity when activated, and this heterogeneity likely arises from differences in the environment surrounding the microglia [26]. Studies have suggested that the caudal and white matter regions of the CNS are more responsive and therefore more vulnerable to inflammatory stimuli [27]. 

Brain expression levels of pro-inflammatory genes were higher in middle-aged compared to young infected mice, while middle-aged mice had similar expression levels of these genes in the systemic compartment. We observed increased mRNA expression of pro-inflammatory mediators HMGB-1, TNF-α, IL-1β, and C1q in the brains of middle-aged infected mice compared to young infected mice. To explore our hypothesis, whereby activated microglia escape their inhibitory mechanisms, we examined specific microglial genes with anti-inflammatory effects [16,17,18], and found a significantly greater expression of CX3CL1, CX3CR1, CD200R and CD47 in middle-aged compared to young infected mice. Based on these results, the hypothesis that the expression of these specific genes is lacking in ageing mice, as an explanation for the greater expression of pro-inflammatory genes, can be rejected.

Microglial activation is associated with the upregulation of TLRs, independently of type of challenge or time point of evaluation [21,28,29,30,31], and the stimulation of TLRs induces a well-characterized signaling cascade, resulting in the transcription of hundreds of pro-inflammatory genes [19]. The TLR signaling cascade can be interrupted at almost every step by negative regulators; for example, by SOCS1 and A20, which block key steps in signal transduction downstream of most TLRs [19]. We found significantly increased mRNA expressions of TLR-2 and MAPK1 in middle-aged infected mice compared to young infected mice, but no notable differences in mRNA expression of SOCS1 or A20. The variety of TLR regulatory mechanisms, present at several checkpoints, indicates that regulation is probably not established by one particular inhibitor alone, and that regulation can be achieved by a cascade of regulators. The increased expressions of TLR-2 and MAPK1 in middle-aged infected mice may suggest that there is a lack of inhibitory regulators of the TLR cascade in the brain of these mice, and this could be responsible for an inappropriate pro-inflammatory response in the central nervous system (CNS) during peripheral infection. Additional studies should be done to reproduce this finding, and future experiments should focus on this TLR signaling cascade and measure other inhibitory aspects of this cascade in the brains of aging mice during peripheral infection. 

Our data show an age-dependent disconnection in the inflammatory transcriptional signature between the brain and systemic compartment. This is in line with previous findings describing that an exaggerated neuro-inflammatory response in aged mice after peripheral LPS challenge was not associated with increased circulating cytokines in the periphery [32]. The age-associated dysregulation of TLR function is seen in peripheral monocytes/macrophages and dendritic cells during LPS stimulation, which can either result in the inappropriate activation or the impairment of TLR signaling [33,34,35]. In mouse brains, physiological aging is associated with the upregulation of TLR-1 to 7, and transcripts of TLR-2 distribute mainly around lateral ventricles close to the choroid plexus at 6 months of age and then gradually extend to the corpus callosum, hippocampus and entire cortex with aging [36]. 

The finding of the greater mRNA expression of several pro-inflammatory mediators in the brain homogenate of middle-aged compared to infected young mice is in line with the literature [12,32,37]. A previous study showed that immunoreactivity for IL-1β was mainly found in the Iba1-positive microglia and partially in neurons, but was rarely found in glial fibrillary acidic protein (GFAP)-positive astrocytes [24], suggesting that these pro-inflammatory mediators are produced by microglial cells. However, at earlier time points, it was shown that TNF-α is transferred to the brain through TNF-α receptors to induce microglial activation, which subsequently produces additional TNF-α and other pro-inflammatory mediators [38]. Increased levels of TNF-α mRNA and protein levels in the brain, liver and serum were observed only 30 min after a single systemic LPS challenge (5 mg/kg). Interestingly, TNF-α levels in the brain remained elevated up to 10 months after LPS injection, while TNF-α levels in the serum and liver normalized after approximately 9 h [38]. We found upregulated mRNA expression of TNF-α in the spleen no earlier than 7 days after *E. coli* injection in both young and middle-aged infected mice, as compared to controls. Moreover, while the mRNA expression of TNF-α was upregulated in the spleen, it was downregulated in the brain at the 7d time point in young infected mice. This indicates once more that the pathogenesis of neuro-inflammation after LPS-induced peripheral inflammation differs from a challenge with live bacteria, and therefore simulates the clinical situation poorly.

Brain autopsies of nine patients with lethal COVID-19 showed an extensive neuro-inflammatory response in all patients, characterized by the activation of microglial cells, affecting both white and grey matter [39]. Immunohistochemically, and with RT-PCR, severe acute respiratory syndrome coronavirus 2 (SARS-CoV-2) was not found in the brain [39], but other researchers have found low concentrations by RT-PCR [40,41]. This suggests SARS-CoV-2 is a non-neuro-invasive virus; however, there is a severe neuro-inflammatory response, and numerous studies shown patients experiencing neurological symptoms [42,43,44]. In the post-illness stage, survivors of severe coronavirus infection experience a depressed mood (11%), insomnia (12%), anxiety (12%), fatigue (19%) and memory impairment (19%) [43]. Experiments have shown the role of systemic NLRP3-inflammasome-interleukin-1β-mediated inflammation, which accounts for the pathological formation of neurofibrillary tangles in murine models of tauopathy, and in this manner can cause neurodegenerative disease [45,46]. This suggests a maladaptive neuro-inflammatory response, and substantiates the evidence of immunomodulation as a target in the treatment of severe COVID-19 and the reduction of the development of neurodegenerative diseases in the following years.

In our search for inhibitory mechanisms that might be lacking in microglial cells during aging, we concentrated on cytokine-triggered pathways in this study. However, neurotransmitters and neurohormones are also known to have pro- and anti-inflammatory influences on microglial cells during inflammation [47]. Microglial cells possess receptors for neurotransmitters and neurohormones such as adenosine triphosphate (ATP), adenosine diphosphate (ADP), glutamate, kianate and neurokinin, which have a pro-inflammatory influence on microglial cells, but also have receptors for gamma-aminobutyric acid (GABA), acetylcholine, norepinephrine, dopamine, bradykinin and cannabinoids, which are known for their neuroprotective and anti-inflammatory influence on microglial cells [47].

There are some limitations to our study. Formally, we cannot differentiate infiltrating myeloid cells and microglial cells using only an Iba-1 staining. We differentiated these cells via their morphology and specific position in the brain parenchyma (around blood vessels). Evidence obtained from preclinical models indicates that the number and phenotype of microglial cells differ between female and male mice [48]. For this study, we decided to use only male mice for three reasons, as follows: (1) to avoid the hormonal cycle of female mice; (2) most other studies, examining microglial cells after systemic inflammatory challenge, used male mice; and (3) because of the exploratory nature of this study, we wanted the mice to differ as little as possible. The results of our study should be reproduced using both male and female mice.

## 5. Conclusions

Our findings indicate an increase in microglial cell number in middle-aged infected mice compared to young infected mice after systemic infection with live *E. coli*. Additionally, in the brains of middle-aged infected mice a more pro-inflammatory environment was found compared to young infected mice. This is in line with our hypothesis postulating an uncontrolled neuro-inflammatory response during aging. In pursuit of a possible explanatory mechanism, we studied inhibitory pathways in microglial cells and negative regulators of the TLR-cascade, and found no evident role for CX3CL3-CX3CR1, CD200-CD200R, CD47-SIRP-α, A20 or SOCS1. The redundancy of negative regulators for microglial cells suggests that regulation might be achieved by a cascade of regulators, indicating that a particular inhibitor might be essential, but not enough to fine-tune signaling, or that different regulators may take action in different brain regions or at different times.

## Figures and Tables

**Figure 1 cells-10-00279-f001:**
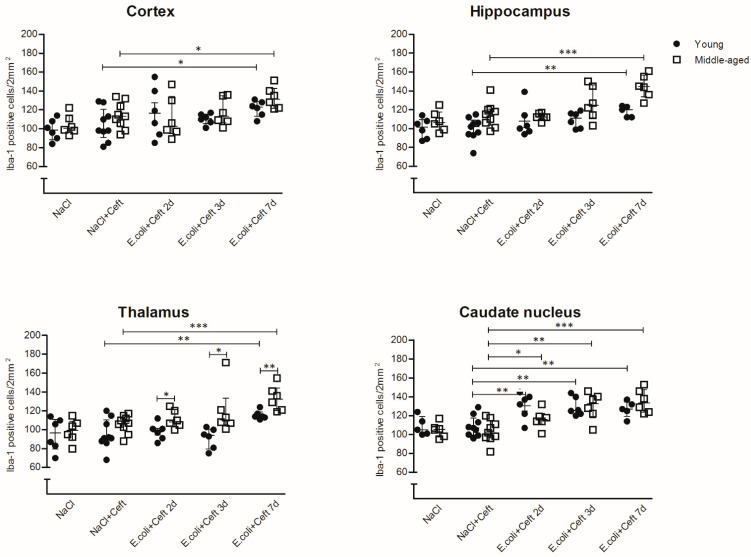
Histopathological counts of microglia. Number of Iba-1-positive microglial cell bodies per experimental group (NaCl: *n* = 6; NaCl + Ceft: *n* = 9; *E. coli* + Ceft 2, 3 and 7d: *n* = 6) in different brain regions (cortex, hippocampus, thalamus and caudate nucleus) per 2 mm^2^. Parametric or non-parametric two-way ANOVA (factors: age and time point) was conducted and continuous variables were tested with post-hoc Mann–Whitney U tests or Student *t*-tests, depending on the distribution of the data. For non-parametric ANOVAs, the ranked transformation of data was performed. Data represent median ± IQR, * *p* ≤ 0.05, ** *p* ≤ 0.01, *** *p* ≤ 0.001.

**Figure 2 cells-10-00279-f002:**
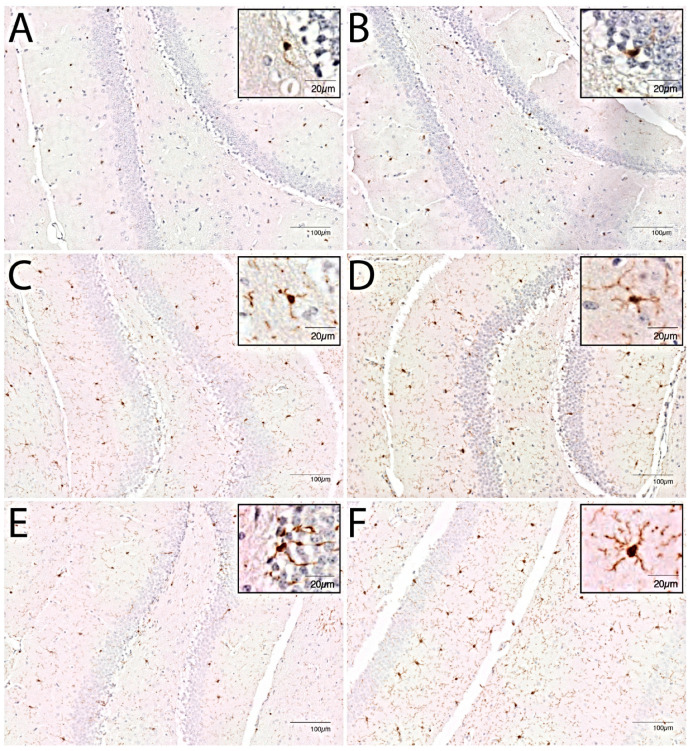
Representative images of immunohistochemical staining with Iba-1 antibody of hippocampus region for young control mice given NaCl and ceftriaxone, sacrificed at day 7 (**A**), middle-aged control mice given NaCl and ceftriaxone, sacrificed at day 7 (**B**), young infected mice given *E. coli* and ceftriaxone, sacrificed at day 3 (**C**), middle-aged infected mice given *E. coli* and ceftriaxone, sacrificed at day 3 (**D**), young infected mice given *E. coli* and ceftriaxone, sacrificed at day 7 (**E**), and middle-aged infected mice given *E. coli* and ceftriaxone, sacrificed at day 7 (**F**).

**Figure 3 cells-10-00279-f003:**
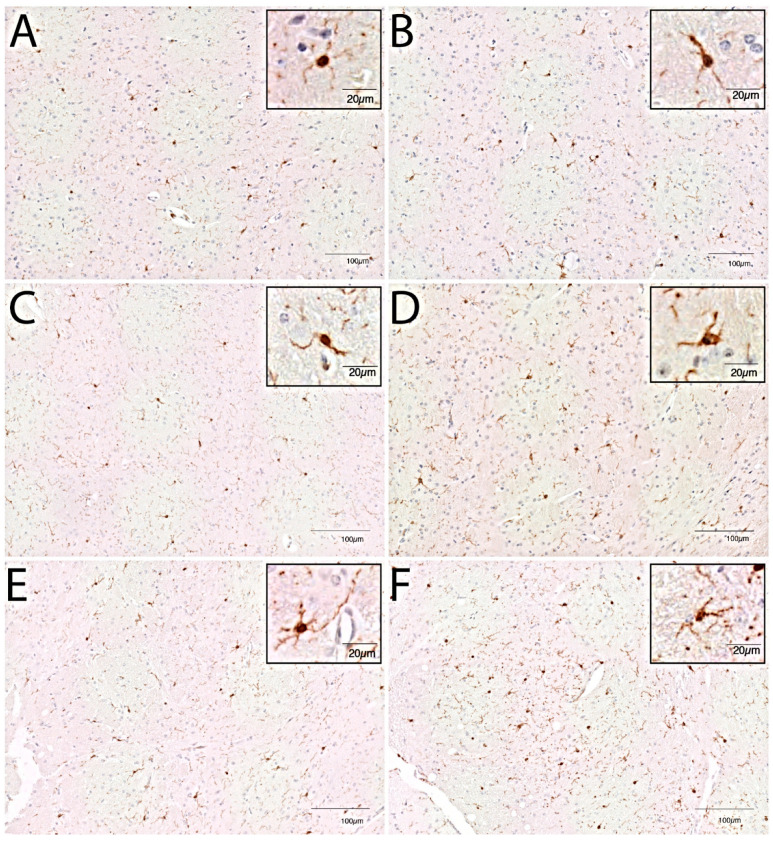
Representative images of immunohistochemical staining with Iba-1 antibody of thalamus region for young control mice given NaCl and ceftriaxone, sacrificed at day 7 (**A**), middle-aged control mice given NaCl and ceftriaxone, sacrificed at day 7 (**B**), young infected mice given *E. coli* and ceftriaxone, sacrificed at day 3 (**C**), middle-aged infected mice given *E. coli* and ceftriaxone, sacrificed at day 3 (**D**), young infected mice given *E. coli* and ceftriaxone, sacrificed at day 7 (**E**), and middle-aged infected mice given *E. coli* and ceftriaxone, sacrificed at day 7 (**F**).

**Figure 4 cells-10-00279-f004:**
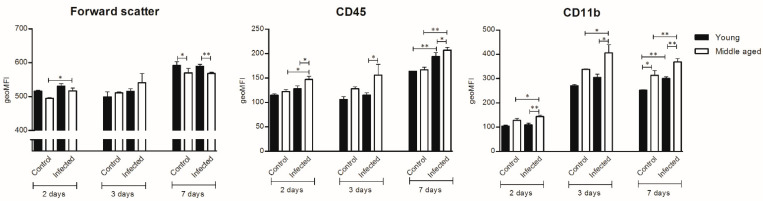
Geometric means (geoMFI) for forward scatter, expression of cluster of differentiation 45 (CD45) and CD11b measured with flow cytometry. Mann–Whitney U tests were performed. Data represent median ± IQR, * *p* ≤ 0.05, ** *p* ≤ 0.01. NB: Flow cytometry for every time point was done on a different day. Laser characteristics vary per day, and therefore the geoMFIs are not comparable between the different time points. As such, every infected group (*n* = 5) has its own control group (*n* = 3 for 2 and 3d time points and *n* = 5 for 7d time point).

**Figure 5 cells-10-00279-f005:**
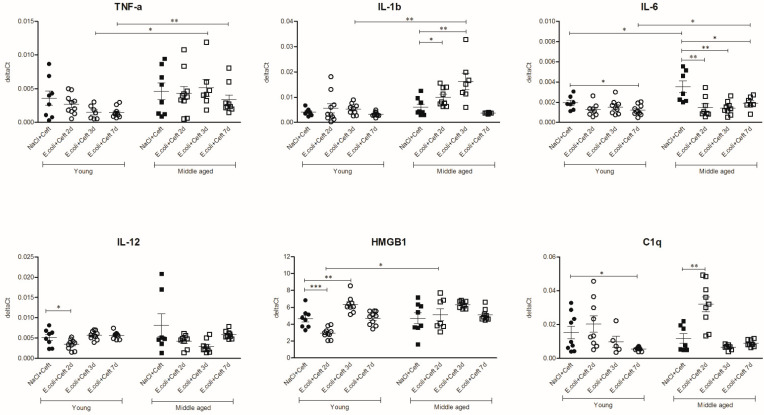
mRNA expression of brain homogenate of general pro-inflammatory mediators TNF-α, IL-1β, IL-6, IL-12, HMGB1 and C1q, illustrated in deltaCts. Group size varies from *n* = 8 to 10. Parametric or non-parametric two-way ANOVA (factors: age and time point) werer conducted and continuous variables were tested with post-hoc Mann–Whitney U tests or Student *t*-tests, depending on the distribution of the data. For non-parametric ANOVAs, the ranked transformation of data was performed. Data represent mean ± SEM, * *p* ≤ 0.05, ** *p* ≤ 0.01, *** *p* ≤ 0.001.

**Figure 6 cells-10-00279-f006:**
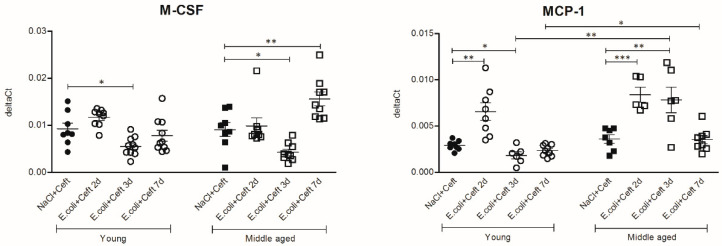
mRNA expressions of brain homogenate of immune regulatory mediators M-CSF and MCP-1, illustrated in deltaCts. Group size varies from *n* = 8 to 10. Parametric or non-parametric two-way ANOVA (factors: age and time point) were conducted and continuous variables were tested with post-hoc Mann–Whitney U tests or Student *t*-tests, depending on the distribution of the data. For non-parametric ANOVAs, ranked transformation of the data was performed. Data represent mean ± SEM, * *p* ≤ 0.05, ** *p* ≤ 0.01, *** *p* ≤ 0.001.

**Figure 7 cells-10-00279-f007:**
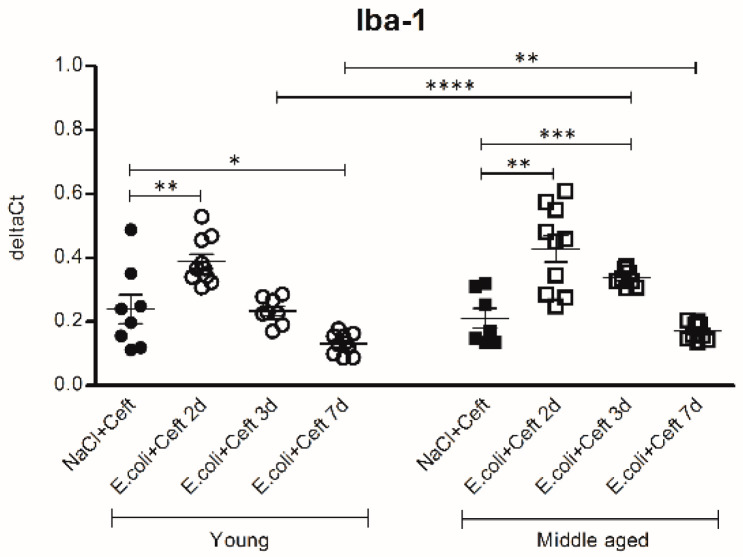
mRNA expression of brain homogenate of myeloid marker Iba-1, illustrated in deltaCts. Group size varies from *n* = 8 to 10. Parametric two-way ANOVA (factors: age and time point) was conducted and continuous variables were tested with post-hoc Student *t*-tests. Data represent mean ± SEM, * *p* ≤ 0.05, ** *p* ≤ 0.01, *** *p* ≤ 0.001, **** *p* ≤ 0.0001.

**Figure 8 cells-10-00279-f008:**
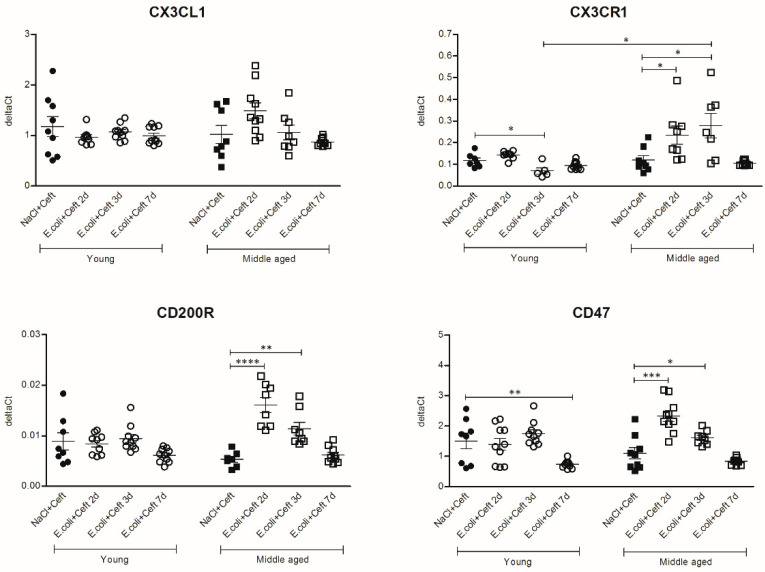
mRNA expression of brain homogenate of specific microglial genes with anti-inflammatory effects: CX3CL1-CX3CR1, CD200R and CD47, illustrated in deltaCts. Group size varies from *n* = 8 to 10. Parametric or non-parametric two-way ANOVA (factors: age and time point) were conducted and continuous variables were tested with post-hoc Mann–Whitney U tests or Student *t*-tests, depending on the distribution of the data. For non-parametric ANOVAs, ranked transformation of the data was performed. Data represent mean ± SEM, * *p* ≤ 0.05, ** *p* ≤ 0.01, *** *p* ≤ 0.001, **** *p* ≤ 0.0001.

**Figure 9 cells-10-00279-f009:**
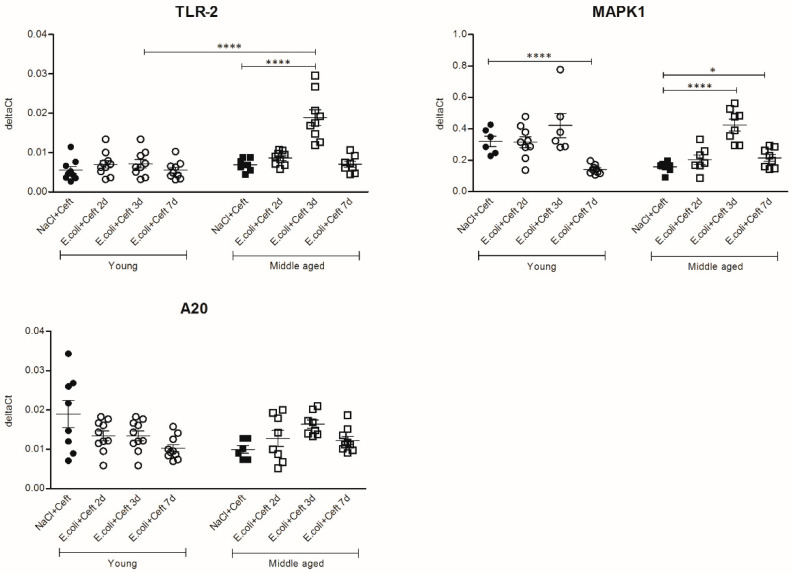
mRNA expression of brain homogenate of components of the TLR signaling cascade: TLR-2, MAKP1 and A20, illustrated in deltaCts. Group size varies from *n* = 8 to 10. Parametric or non-parametric two-way ANOVA (factors: age and time point) were conducted and continuous variables were tested with post-hoc Mann–Whitney U tests or Student *t*-tests, depending on the distribution of the data. For non-parametric ANOVAs, ranked transformation of the data was performed. Data represent mean ± SEM, * *p* ≤ 0.05, **** *p* ≤ 0.0001.

**Figure 10 cells-10-00279-f010:**
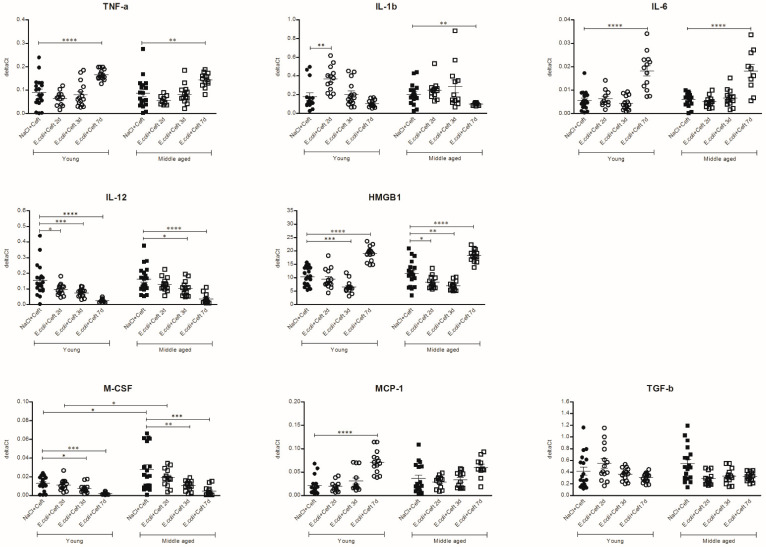
mRNA expression of spleen homogenates of TNF-α, IL-1β, IL-6, IL-12, HMGB1, M-CSF, MCP-1, and TGF-β, illustrated in deltaCts. Group size varies from *n* = 10 to 15. Parametric or non-parametric two-way ANOVA (factors: age and time point) were conducted and continuous variables were tested with post-hoc Mann–Whitney U tests or Student *t*-tests, depending on the distribution of the data. For non-parametric ANOVAs, ranked transformation of the data was performed. Data represent mean ± SEM, * *p* ≤ 0.05, ** *p* ≤ 0.01, *** *p* ≤ 0.001, **** *p* ≤ 0.0001.

## Data Availability

Data are contained within the article or supplementary material.

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
