# Peer review of "Aging and Microglial Response following Systemic Stimulation with Escherichia coli in Mice"

_cells, 2021, doi:10.3390/cells10020279_

Round 1

Reviewer 1 Report

This is a study about central inflammation after peripheral E coli exposure. A lot of animals were sacrificed and the data should be published.

The paper itself can be improved. The Abstract should be improved so that it is more reader friendly, e.g. remove text about delirium (has nothing to do with your study), explain what ceftriaxone is as soon as you use the term in the abstract. Also define "young" and "middle aged". The reader should be able to understand what you have done by simply reading the abstract, rather than having to google certain terms while reading your abstract, or searching over the methods. Futhermore, 2 year old mice are not middle-aged, they are old. Mice in a laboratory often die of natural causes by this age. Rather than "young" or "middle-aged" you can simply put the age, it will not affect the data or conclusions of your paper anyway. You also refer to them as "Old" in your Appendix A.

Data presentation could be nicer, for example Fig 1 has these thick lines as axes and error bars and mean, and very thin lines as "Middle aged" data points. Use the same thickness for all your lines/borders otherwise it is painful to look at.

Always indicate the statistical test performed in every figure.

Figure 2,3: the scale bars are not readable and extremely blurry. There is no reason for this, also only one scale bar is necessary per figure, rather than 6.

Figure 4, you have used multiple fonts, i.e. thick for the x axis but thin for the "2 days", "3 days", "7 days" below it. It is painful to look at and distracting, and furthermore gives the impression that you did not spend a lot of time with your data, and did not aim for a perfect presentation. Rather, it appears that the figures were put together hastily, and this gives and un-serious impression which not only makes your paper look un-serious, it also reflects poorly on the journal that will accept your work. The lines for statistical analysis are also very thin for no reason.

Figure5A: the lines for statistical analysis are very difficult to follow, as it is unclear what scatter plots are being compared. The lines are hanging in the air and going down all the way to the bars compared. Please extend the vertical lines so that they clearly indicate which bars/scatter plots are being compared. Always mention the statistical test performed in the figure.

Figure 5D: Calling some microglial genes "microglial activation inhibitors" is not acceptable. These are not inhibitors, they are just genes. Those papers you refer to describe them in some context under defined conditions in which they may have anti-inflammatory effects. That does not make them microglial inhibitors. It only makes them genes. A microglial inhibitors can be a molecular compound or peptide, but not a gene. This has to be re-written.

Author Response

We would like to thank you for your valuable comments. Please find our point to point response below.

1. The abstract should be improved so that it is more reader friendly. Remove text about delirium (has nothing to do with your study), explain what ceftriaxone is as soon as you use the term in the abstract. Also define "young" and "middle aged".

Reply: we agree, the suggested changes are made in the abstract (lines 17-18, 21 and 23).

2. 2 year old mice are not middle aged, they are old

Reply: 18-24 month old mice are typically used in studying aging and defined as old. Our mice are 13 to 14 months old and therefore considered as adolescent or middle-aged.

3. Data presentation could be nicer. Always mention the statistical test performed in the figure

Reply: we received more comments on the figures from the reviewers. To exclude that this was due to a conversion problem I have converted all figures to TIF files and also made sure all the axes, error-bars, means and lines for age and days are the same thickness. Statistical test information is added to the figure legends (lines 328-330, 351, 385-387, 410-412, 433-435, 468-470, 507-509,  531-533, 718, 734 and 737-740).

4. Figure 2 an 3: the scale bars are not readable and extremely blurry

Reply: this was probably also due to a conversion problem, I converted the original files to TIF files.

5. Calling microglial genes “microglial activation inhibitors” is not acceptable. These are not inhibitors, they are just genes. This has to be re-written

Reply: we agree, this has been re-written (Lines 242-243, 437, Legend of Figure 8 line 467 and 574-575)

Reviewer 2 Report

In this manuscript, the authors investigated the effects of the systemic challenge with live bacteria and the role of age on the neuro-inflammatory response. They used animal models to do this. Overall, they observed significant differences in microglial cell number and inflammatory response between young and middle-aged mice.

The topic of this paper is clear and of interest to the reader. Overall, data support the authors' conclusions.

The authors used young and middle-aged mice to define the role of ageing, but I wonder if specific ageing markers are for instance differentially expressed in these mice. This should be investigated.

Author Response

We would like to thank you for your valuable comments. Please find our response below.

The authors used young and middle-aged mice to define the role of ageing, but I wonder if specific ageing markers are for instance differentially expressed in these mice. This should be investigated.

Reply: This is an interesting topic and a fascinating way to define and investigate ageing. As the reviewer mentioned, for this study we used the age of the mice and did not examine specific ageing markers. This should be included in future studies.  

Reviewer 3 Report

In this manuscript (cells-1082529), the authors demonstrated that systemic challenge with live bacteria causes an age-dependent neuroinflammatory and microglial response. Some concerns and suggestions are listed as below:

  1. In Figure 1, the number of Iba-1 positive microglia were counted in different brain regions. However, Ki-67, a proliferation marker, should also be stained.
  2. The authors indicated an increase in microglial cell number in middle-aged infected mice compared to young infected mice after systemic infection with live E. coli. I suggest that F4/80 should also be used. In figure 1, it is not clear for readers if the increased numbers of cells are resident microglia or infiltrating myeloid cells.
  3. The current evidence obtained from preclinical models indicates that the number and phenotype of microglia differ between females and males in a region- and age-specific manner (Transcriptional and Translational Differences of Microglia from Male and Female Brains. Cell Rep 2018; Sex-Specific Features of Microglia from Adult Mice. Cell Rep 2018). In addition, an underestimated yet marked sex-dependent microglial activation pattern may exist in the injured CNS (Sex-Specific Effects of Microglia-Like Cell Engraftment during Experimental Autoimmune Encephalomyelitis, International Journal of Molecular Sciences, 2020). This point should be discussed. Importantly, the authors should use both male and female mice.
  4. Aging itself may change microglial density, distribution, and ramified morphology. Did the authors observe any microglial differences (number or morphology) between young and middle aged mice?
  5. The authors should also measure inflammatory changes in the circulation following systemic stimulation with Escherichia coli.
  6. In lines 290-293, figures 2-3 should be introduced before figure 4. Please double check.
  7. The scale bars are missing in both figure 2F and figure 3F.
  8. In figure 4, the authors should gate and then measure CD11b+CD45low resident microglia and CD11b+CD45hi infiltrating macrophages, respectively.
  9. Figure 5A-E should be combined into one figure. Or use figure 5, 6, 7…
  10. In Figure 5A, why the cytokine production was not measured in a brain-specific manner?
  11. In Figure 5C, there are some unnecessary words (numbers) in the figure. Please double check. The same mistake was also found in figure 5D.
  12. Apart from microglia, astrocytes may also be altered following systemic challenge with live bacteria.
  13. In this study, the authors did not distinct parenchymal and perivascular macrophages in the brain. A recent study showed that border associated macrophages display a response to a peripheral endotoxin challenge distinct from microglia (A Binary Cre Transgenic Approach Dissects Microglia and CNS Border-Associated Macrophages, Immunity, 2020)
  14. Peripheral immune stimulation evokes immune memory in microglia and then shapes neuropathology (Innate immune memory in the brain shapes neurological disease hallmarks, Nature, 2018). I wonder if different times of intraperitoneal injections following systemic challenge with live bacteria (low dose) may cause different brain responses.

Author Response

We would like to thank you for your valuable comments. Please find our point to point response below.

1. In Figure 1, the number of Iba-1 positive microglia were counted in different brain regions. However, Ki-67, a proliferation marker, should also be stained.

2. The authors indicated an increase in microglial cell number in middle-aged infected mice compared to young infected mice after systemic infection with live E. coli. I suggest that F4/80 should also be used. In figure 1, it is not clear for readers if the increased numbers of cells are resident microglia or infiltrating myeloid cells.

Reply to both comments: we agree. Formally we can’t differentiate infiltrating myeloid cells and microglial cells with only an Iba-1 staining. We differentiated these cells by morphology and specific position in the brain parenchyma (around blood vessels). We use the term microglial cell number and not microglial proliferation, because we are aware we didn’t use a proliferation marker. We added this issue in the discussion (Lines 653-656).

3. The current evidence obtained from preclinical models indicates that the number and phenotype of microglia differ between females and males in a region- and age-specific manner (Transcriptional and Translational Differences of Microglia from Male and Female Brains. Cell Rep 2018; Sex-Specific Features of Microglia from Adult Mice. Cell Rep 2018). In addition, an underestimated yet marked sex-dependent microglial activation pattern may exist in the injured CNS (Sex-Specific Effects of Microglia-Like Cell Engraftment during Experimental Autoimmune Encephalomyelitis, International Journal of Molecular Sciences, 2020). This point should be discussed. Importantly, the authors should use both male and female mice.

Reply: we agree that this is an important topic and included this in the discussion (line 656-662).

We decided to use only male mice to avoid the hormonal cycle of female mice, knowing that this most likely will affect the immune system. Because the exploratory nature of this study we wanted mice to differ as little as possible.

Furthermore, most other studies, examining microglial cells after systemic inflammatory challenge, used male mice (Systemic inflammation and microglial activation: systemic review of animal experiments, Journal of neuroinflammation, 2015).

4. Aging itself may change microglial density, distribution, and ramified morphology. Did the authors observe any microglial differences (number or morphology) between young and middle aged mice?

Reply: in control mice injected with only saline and in control mice injected with saline and ceftriaxone, we did not find differences in number or morphology between young and middle-aged mice in the different brain regions. This is indeed an important subject and we added this in the manuscript (line  314-315).

5. The authors should also measure inflammatory changes in the circulation following systemic stimulation with Escherichia coli.

Reply: we agree. Unfortunately, we weren’t able to draw enough blood from all mice to do measure all the cytokines. Nevertheless, we think that mRNA expression in spleen represents the systemic compartment.

6. In lines 290-293, figures 2-3 should be introduced before figure 4. Please double check.

Reply: we agree, has been checked and solved.

7. The scale bars are missing in both figure 2F and figure 3F.

Reply: we received more comments on the figures from the reviewers. To exclude that this was due to a conversion problem I have converted all figures to TIF files.

8. In figure 4, the authors should gate and then measure CD11b+CD45low resident microglia and CD11b+CD45hi infiltrating macrophages, respectively.

Reply: we did, but we did not find a lot of CD11b+CD45hi cells during flowcytometry after cerebral perfusion with sterile PBS during termination (see figure A1 and A2 of the appendix)

9. Figure 5A-E should be combined into one figure. Or use figure 5, 6, 7…

Reply: we agree, we changed this. Figure 5A is 5, 5B is 6, 5C is 7, 5D is 8, 5E is 9, and figure 6 is now 10.

10. In Figure 5A, why the cytokine production was not measured in a brain-specific manner?

Reply: we tried to do this for several cytokines with Luminex multiplex assays but had problems with the lower limit of detection and the necessary dilution steps of the brain homogenate.

11. In Figure 5C, there are some unnecessary words (numbers) in the figure. Please double check. The same mistake was also found in figure 5D.

Reply: see reply at point 7, I have converted all figures to TIF files.

12. Apart from microglia, astrocytes may also be altered following systemic challenge with live bacteria.

Reply: this is most definitely true. Astrocytes have important homeostatic tasks, including the formation of barriers protecting the brain from peripheral influences. In essence astrocytes form the nervous tissue that constitutes the blood brain barrier (BBB) and regulate the permeability of the BBB through influencing expression of tight junctions in endothelial cell layers. However this was beyond the scope of this study. It would be very interesting to examine astrocyte function in future experiments using live bacteria, to better understand why neuro-inflammation arises while the pathogen itself does not seem to invade the central nervous system.

13. In this study, the authors did not distinct parenchymal and perivascular macrophages in the brain. A recent study showed that border associated macrophages display a response to a peripheral endotoxin challenge distinct from microglia (A Binary Cre Transgenic Approach Dissects Microglia and CNS Border-Associated Macrophages, Immunity, 2020)

Reply: see also reply at point 1 and 2 and the added section in the discussion (Lines 653-656).

14. Peripheral immune stimulation evokes immune memory in microglia and then shapes neuropathology (Innate immune memory in the brain shapes neurological disease hallmarks, Nature, 2018). I wonder if different times of intraperitoneal injections following systemic challenge with live bacteria (low dose) may cause different brain responses.

Reply: this is an interesting thought. There are several studies which used multiple hit models with LPS (see table 2 in Systemic inflammation and microglial activation: systemic review of animal experiments, Journal of neuroinflammation, 2015). It also has been shown that repeated dosing with LPS induces features of neurodegenerative disease (Cunningham, Microglia an neurodegeneration: the role of systemic inflammation, Glia 2012). To my knowledge there are no multiple hit studies using live bacteria.

Reviewer 4 Report

General comments

The authors touch a question of high clinical importance. Hitherto, the diagnosis delirium is inflationary used, especially in geriatric units. However, delirium is not really defined -rather used as a puzzle of symptoms. Actually, these symptoms can be due to various origins. The employment of the possibilities of therapeutic access needs the identification of the mechanisms of occurrence of such disturbances of brain function which, without doubt, can be found also also in peripheral and systemic triggers like infections including, at the moment, also COVID 19 , sepsis  and also drugs or drug cocktails. 

The authors develop an ambitious approach in an established mouse model after i.p. application of E.coli and the antibiotic ceftriaxone. The model allows observation of inflammatory and immune response in the brain and spleen through seven days. Cerebral microglial cell number is investigated in cortex, nucleus caudatus, hippocampus and thalamus by cytofluorometry. mRNA expression of pro-inflammatory and immune-regulatory mediators as well as components of the toll-like receptor cascade are investigated in homogenates of the left brain and in homogenates of the spleen. Additionally, morphological changes of cerebral microglia in the right hemisphere are compared at the end of the experiments.

Influence of age and infection on mRNA expression measured in homogenates of the left hemisphere of the brain, can be observed, especially, in middle-aged mice (13-14 month old) compared to young (2 month old) animals.

The authors find increases in middle-aged mice compared to young  for pro-inflammatory markers, markers of microglial activation, and components of the  toll-like receptor cascade . The authors emphasize different time courses in the changes of biomarkers in middle-aged animals in comparison to young,  emphasize the absence of a role of evident actions of negative regulators of the toll-like cascade. Moreover, they presume a disconnection of the transcriptional signature between the brain and the systemic compartment.

 The authors observe an increased number of microglial cells in several brain regions of middle-aged animals compared to young animals. Moderate morphological differences between microglial cells in young and middle-aged animals are demonstrated in thalamus and hippocampus.

The description of methods has been done carefully and detailed, also with supplementary data. However, the presentation of the results suffers from some confusing  conditions different from that announced in methods.

Major remarks

With view on the relatively low number of investigated animals, especially in some control groups, and on the fact that it is known  in vivo experiments show usually  a rather higher standard deviation and the in vivo methods higher relative standard deviations than in vitro methods, the conclusion of generally uncontrolled response (line 592) of the middle-aged animals  seems too early. Perhaps the authors should reduce the hypothesis to a role of the investigated parameters in the disturbance of the immunological  and inflammatory  response. Especially, if the authors recognize in their own comments that it has to be shown that the results are reproducible and that there are a lot of further factors which could be involved (line 663-668).

Minor remarks

  • Line 105, lines 132-135 , Appendix table A1: In methods, the size for the control groups is announced with 6-8 animals per time point in each group. In results, the sizes of the groups shift between 3 (e.g. in controls of brain samples) and 9 (brain) and 15 (spleen). And even in methods (lines 132-135) are shown different control group sizes (5-10 animals). Also, there is not clear, whether the group sizes in experimental groups are valuable for brain, spleen or liver. That is confusing and rises also questions on the choice of the statistical methods. Even in legend of table A1 (line 711) is not clear whether these groups are used for investigation of brain or of spleen.
  • Line 129: was the liver objective of investigations in the study? There are no results?
  • Line 284: Better to write : ..show continuously increased cell number…

        (the increase seems not to be continued than rather to keep a higher 

         level)

  • Line 354, Figure 4: could the authors give a RSD for flow cytometric control investigations ?
  • Line 354, Figure 4: Why the authors decided for the geoMFI (normally secondary choice) instead of MFI.
  • Line 447, line 478: Figure 5c and 5d ; Appendix, Figures A4, A5 :in this figures is not clear what are the additional big cyphers in the diagrams?
  • It would be useful if the authors would add a table with description of physiological roles of the biomarkers determined.
  • The authors describe alterations of CX3XL and CX3CR1, CD200 etc. (lines 679-680) but conclude that there is no evident role. Can that be excluded really by the data shown. Please explain.
  • Please check for typos:

               Line 184:  presence instead of presents

                Line 222: Spectrophotometer

                Line 238 and line 676: Additionally, we examined

               Line 242:   Lastly, we examine

               Line 648 : ……was not found in  the brain…

Author Response

We would like to thank you for your valuable comments. Please find our point to point response below.

Major remarks

With view on the relatively low number of investigated animals, especially in some control groups, and on the fact that it is known in vivo experiments show usually a rather higher standard deviation and the in vivo methods higher relative standard deviations than in vitro methods, the conclusion of generally uncontrolled response (line 592) of the middle-aged animals seems too early. Perhaps the authors should reduce the hypothesis to a role of the investigated parameters in the disturbance of the immunological  and inflammatory  response. Especially, if the authors recognize in their own comments that it has to be shown that the results are reproducible and that there are a lot of further factors which could be involved (line 663-668).

Reply: we fully agree with your statement. Our hypothesis is the leading question of a bigger research project, where this specific study is a small part of. We cannot conclude that the results of this study confirm our hypothesis but our results are directorial to a more pro-inflammatory state in middle-aged mice. Therefore we rephrased  line 542-543 and removed “fuelling a chronic uncontrolled neuro-inflammation” in line 574.

Minor remarks

1. Line 105, lines 132-135 , Appendix table A1: In methods, the size for the control groups is announced with 6-8 animals per time point in each group. In results, the sizes of the groups shift between 3 (e.g. in controls of brain samples) and 9 (brain) and 15 (spleen). And even in methods (lines 132-135) are shown different control group sizes (5-10 animals). Also, there is not clear, whether the group sizes in experimental groups are valuable for brain, spleen or liver. That is confusing and rises also questions on the choice of the statistical methods. Even in legend of table A1 (line 711) is not clear whether these groups are used for investigation of brain or of spleen.

Reply: we agree that this is confusing. To avoid this confusion we made changes in the text to make it more clear, line 131-141, and referred to the figures for the exact group sized, line 144-146.

To clarify, in round 1 half of the brain was used for flowcytometry, the other half for immunohistochemistry or cytokines measurements of brain homogenate. In round 2 half of the brain was used for mRNA expression and bacterial outgrowth measurements, the other half for immunohistochemistry or cytokines measurements of brain homogenate. Beforehand, we randomly chose which half of the brain was used for which analysis.

We tried to measure several cytokines in brain homogenate with Luminex multiplex assays, but failed to do so due to technical issues. We had problems with the lower limit of detection and the necessary dilution steps of the brain homogenate.

The main reason to choose to divide flowcytometry and mRNA expression in two different rounds was the daily variety of laser characteristics for flowcytometry. This could have been solved using other statistical tests, but after power calculations we would have needed to use all lot more animals.

For spleen the sizes of the groups are bigger because we only measured mRNA expression and bacterial outgrowth.

2. Line 129: was the liver objective of investigations in the study? There are no results?

Reply: Liver was only used for bacterial outgrowth measurements (line 141-143 and Appendix, Figure A4)

3. Line 284: Better to write : ..show continuously increased cell number…(the increase seems not to be continued than rather to keep a higher level)

Reply: revised as requested (Line 287).

4. Line 354, Figure 4: could the authors give a RSD for flow cytometric control investigations?

Reply: robust standard deviations of control groups are stated below. 

Young control

Middle aged control

Forward scatter

Day 2

10

5

Day 3

12

3

Day 7

8

12

CD45

Day 2

5

5

Day 3

6

4

Day 7

2

12

CD11b

Day 2

4

5

Day 3

13

13

Day 7

3

13

5. Line 354, Figure 4: Why the authors decided for the geoMFI (normally secondary choice) instead of MFI.

Reply: Since fluorescence intensities increase logarithmically, the geometric MFI indicates the population centre better than the arithmetic MFI. This is true if the fluorescence intensities are distributed in a Log-normal manner, but also in other cases, for example because the geometric mean is not affected as strongly as the arithmetic mean by small numbers of outliers, or by skewing of the tails of a distribution. So, in short, we used the geometric MFI, since it is a more robust measure of overall fluorescence intensity that the arithmetic mean.

6. Line 447, line 478: Figure 5c and 5d ; Appendix, Figures A4, A5 :in this figures is not clear what are the additional big cyphers in the diagrams?

Reply: we received a lot of comments on the figures from all reviewers. To exclude that this was due to a conversion problem I have converted all figures to TIF files

7. It would be useful if the authors would add a table with description of physiological roles of the biomarkers determined.

Reply: Although a good suggestion, we tried to cluster the biomarkers to function in the results and tried to clarify the biomarkers that drive our findings in the discussion.

8. The authors describe alterations of CX3XL and CX3CR1, CD200 etc. (lines 679-680) but conclude that there is no evident role. Can that be excluded really by the data shown. Please explain.

Reply: we agree, we revised  this text (Line 577-579).

CX3CL1, CX3CR1, CD200R and CD47 are specific microglial genes with anti-inflammatory features. Our hypothesis is that older mice lack anti-inflammatory cascades and therefore show a more pro-inflammatory milieu in the brain. Our results roughly show more expression of these genes, with anti-inflammatory features, in middle-ages animals.  So the hypothesis that the expression of genes is upregulated in young mice and lacking in ageing mice is not true. This is the only conclusion which can be made with this data.

9. Please check for typos:

- Line 184:  presence instead of presents, line 187

- Line 222: Spectrophotometer, line 225

- Line 238 and line 676: Additionally, we examined, line 241 and line 667

- Line 242:   Lastly, we examine, line 245

- Line 648 : ……was not found in  the brain… line 630-631

Reply: all typos are corrected

Round 2

Reviewer 2 Report

The manuscript is suitable for the publication.

Reviewer 3 Report

The authors did not find any differences in number or morphology between young and aged mice in different brain regions. This is not consisent with previous literature. Please discuss this in the discussion.

The authors did not find CD11b+CD45hi infiltrating macrophages following systemic stimulation with Escherichia coli. I do not know why? Please provide more evidence regarding systemic inflammation and neuroinflammation.